# The Effect of Inhibin Immunization in Seminiferous Epithelium of Yangzhou Goose Ganders: A Histological Study

**DOI:** 10.3390/ani11102801

**Published:** 2021-09-26

**Authors:** Muhammad Faheem Akhtar, Ejaz Ahmad, Ilyas Ali, Muhammad Shafiq, Zhe Chen

**Affiliations:** 1Jiangsu Province Key Laboratory for Molecular and Medical Biotechnology, College of Life Science, Nanjing Normal University, Nanjing 210023, China; 90741@njnu.edu.cn; 2Department of Clinical Sciences, Faculty of Veterinary Sciences, Bahauddin Zakariya University, Multan 60800, Pakistan; ejazvetrohail@gmail.com; 3College of Animal Science and Technology, Nanjing Agricultural University, Nanjing 210095, China; ilyasnjau@gmail.com; 4Department of Cell Biology and Genetics, Shantou University Medical College, Shantou 515063, China; drshafiqnjau@yahoo.com; 5Key Laboratory of Crop and Livestock Integration, Ministry of Agriculture, Institute of Animal Science, Jiangsu Academy of Agricultural Sciences, Nanjing 210014, China

**Keywords:** inhibin immunization, Yangzhou goose ganders, seminiferous epithelium

## Abstract

**Simple Summary:**

Achieving optimum fertility is the ultimate goal of keeping gander flock in commercial geese farming. Being seasonal breeders, geese exhibit poor reproductive efficiency. Inhibin (INH) immunization has improved reproductive parameters in some mammalian species but remained obscure in birds, especially geese. The present study aimed to check the effect of INH immunization on testicular histoarchitecture of Yangzhou ganders. Results illustrated that INH immunization disrupted seminiferous epithelium, germ cells development, lowered efficiency of spermatogenesis, and caused apoptosis in seminiferous tubules of Yangzhou ganders.

**Abstract:**

The current study investigated the effect of inhibin immunization on germ cell numbers (spermatogonia, spermatocytes, round, and elongated spermatids), seminiferous tubules (ST) diameter, Johnsen’s score, epithelial height (μm), luminal tubular diameter (μm), and number of ST per field (ST/field) of Yangzhou goose ganders. Histological evaluation showed apoptosis and regression of testes after inhibin (INH) immunization, with a concomitantly marked reduction in the round and elongated spermatids in the experiment (INH) group compared to the control group. The diameter of seminiferous tubules (ST) and epithelial height (EH) were positively correlated at 181, 200, and 227 days of age. In comparison, luminal tubular diameter (LD) was negatively correlated on day 227 to ST diameter and epithelial height. On day 227, many seminiferous tubules per field (ST/field) were negatively correlated to ST diameter, EH, and LD. INH immunization elevated ST diameter, EH, and LD, while Johnsen’s score and number of ST/field had reciprocal expression. In conclusion, the concomitant effect of INH immunization and seasonality in breeding regressed germ cells and damaged spermatogenesis in seminiferous epithelium Yangzhou ganders.

## 1. Introduction

China has the most robust goose industry, with an annual output accounting for roughly 95% of the world’s goose production: 600 million geese slaughtered, producing 250 million tons of meat [1]. However, low reproductive efficiency remains a hurdle in achieving a holistic production approach, including seasonality in breeding. Inhibin (INH) exhibits a vital role on the hypothalamus-pituitary gonadal (HPG) axis [2]. In males, INH is secreted by Sertoli cells of testes, and in females, granulosa cells of ovarian follicles [3]. INH is a glycoprotein having a molecular weight of 31–34 kDa, forming a disulphide-linked dimer that shares a common α-subunit and differs in β-subunit (βA-subunit and βB-subunit), βA in INH A (α-βA), and βB in INH B (α-βA). INH belongs to the transforming growth factor (TGF-β) superfamily and has been proposed as an autocrine/paracrine factor that regulates follicular atresia, growth, steroidogenesis, and gonadotropin responsiveness [4]. INH is a negative feedback regulator of Follicle-stimulating Hormone (FSH) that directly acts on Sertoli cells in males. Sertoli cells are central regulators of testes development [5]. Germ cells are embedded inside Sertoli cells that provide protection and nourishment to developing germ cells [6]. Immunization against INH has been used in avian and mammalian species, including seasonal breeders, to improve reproductive efficiency and spermatogenesis [7,8,9,10]. Active immunization against INH α-subunit has been used as a method to check the physiology of INH in bulls and rams [11].

INH inoculation elevates the semen quality (fresh and post-thaw) in Beetal goat bucks during low or peak breeding seasons [12]. In female Japanese quails, recombinant chicken INH protein immunization enhanced egg production and puberty [7]. INH immunization elevated hypothalamic GnRH-I mRNA expression in ageing white leg-horn roosters [8] and important reproductive function in young leghorn roosters [9]. In Patridge Shank hens, INH vaccine pcISI can influence antibody production against inhibin and enhance follicle development and egg-laying performance [13]. In our previous study, active immunization against INH effected spermatogenesis and testicular development by expressing hypothalamic, pituitary, and testicular genes in Yangzhou goose ganders [10]. The relationship between gonadal derived INH and FSH are well documented in mammalian species but remains devoid in avian species, particularly in ganders. The Yangzhou goose is a long-day breeding bird, starts egg-laying in autumn, peaks between February and March, and ends between May and June [14]. Similar to other seasonal breeders, goose breeds in China vary in breeding seasonality based on locations [1]. Due to solid seasonality in breeding, ganders have lower testosterone concentrations and semen quality, specifically in non-breeding seasons. The present study elucidated changes in the seminiferous epithelium, germ cells variation, and efficiency of spermatogenesis in Yangzhou goose ganders after INH immunization.

## 2. Materials and Methods

### 2.1. Ethics Statement

Experimental protocols were conducted in light of the Guide for Care and Use of Laboratory Animals approved by the institutional Animal Care and Use Committee of Nanjing Agricultural University, China (Approval Numbers: 31572403 and 31402075).

### 2.2. Birds and Management

In April–June, the experiment was performed at Sunlake swan farm, Henglin Township, Changzhou, Jiangsu, China. Birds were individually identified using tags placed through their inner wings to distinguish them from other birds and avoid pecking. Until the end of the experiment, ganders were kept at ambient temperatures between 25 °C and 32 °C. Birds had free access to drinking water and were fed ad libitum with a 12.5% crude protein mix, supplemented with green grass whenever possible. During the daytime, the feed was offered. Until the end of the experiment, the lighting schedule was 11 L:13 D.

### 2.3. Immunogen Preparation

For the expression of goose inhibin (INH) fusion protein, the cDNA sequence of mature inhibin peptide was inserted into the BamHI and HindIII sites of the expression vector pRSET A and then transfected the reconstructed vector into bacteria strain *E. coli* BL21 (DE3). This recombinant protein contained 149 amino acid residues, including a 36-residue leading sequence derived from the expression plasmid pRSET A, apart from the 113-residue sequence of goose inhibin α-subunit mature peptide. This recombinant protein was purified and homogenized with mineral oil adjuvant (Solarbio Life Sciences, Nanjing, China) to reach a final concentration of 1 mg/mL for use in the experiment as immunogen. The bovine serum albumin (BSA) was prepared with physiological saline homogenized with mineral oil adjuvant.

### 2.4. Experimental Design

A flock of Yangzhou ganders (*n* = 60) at 161 days of age, body weight 4.70 kg, having the same genetic origin, were equally assigned in groups A and B. In Yangzhou ganders, body maturity is different from sexual maturity, i.e., the body matures at 161 days of age while sexual maturity is achieved at 227 days of age [15]. At 161 days of age, ganders in group B were inoculated against INH (1 mg/mL Inhibin; I/M) followed by booster immunizations of INH at 181 and 209 days. Yangzhou ganders in group A received intramuscular BSA injection and served as the control group.

### 2.5. Tissue Collection

Testis tissues were collected at 181, 200, and 227 days of age, as there exists a difference in testis maturation and body maturation in Yangzhou ganders. Initially, on 181 days of age, ten ganders from both groups were slaughtered for testes samples. Similarly, ten ganders from both groups were slaughtered at 200 and 227 days of age. Birds were killed by cervical dislocation during the experiment. Immediately after slaughtering, testes tissues were frozen in liquid nitrogen and stored at −80 °C.

### 2.6. Microscopy

Testes tissues were cut perpendicular to the longest axis at 5 μm and mounted on glass. A small piece of left testes tissue (0.125 cm^3^) from each bird was taken, fixed in 10% buffered neutral formalin solution for 24 h, and used for histological evaluation using an automated tissue processor (LEICA RM 2235). Then, tissues were dehydrated in alcohol of increasing concentrations, i.e., 70%, 80%, 90%, and 100% absolute alcohol, cleared in xylene, and embedded in molten paraffin wax. Tissue staining was performed using hematoxylin and eosin (Nanjing Jiancheng Bioengineering Institute, Nanjing, China). To observe changes in ST, the testes tissue stained slides were observed individually under bright field Olympus BX63 light microscope (OLYMPUSBX63,Olympus Corporation, Tokyo, Japan) at 10x and 40x magnification. The photomicrographs had been obtained using an 8 MegaPixel CCD digital camera procedure fitted to the microscope. The mean of two opposite measurements, EH1 and EH2, was taken to calculate epithelial height (EH). The total diameter of ST was subtracted from epithelial height (EH1 + EH2) to calculate luminal tubular diameter (LD). Histological fields were determined from stained slides using the Image J public domain software [16]

Germ cells development in the ST was classified according to the Johnsen score. It applies for the numbers from 1 to 10 to a cross-section of each tubule according to the following criteria:10 = complete spermatogenesis;9 = spermatozoa present with random appeared spermatogenesis;8 = few spermatozoa;7 = no spermatozoa, but spermatids present;6 = few spermatids present;5 = only spermatocytes present;4 = few spermatocytes present;3 = spermatogonia present;2 = only Sertoli cells present; and1 = almost empty lumen.

The mean score was calculated by randomly selecting 10 ST/birds.

### 2.7. Statistical Analysis

We used SPSS (Version 20.0) Newyork, USA and Graph Pad Prism, California, USA (Version 5.0) for statistical analysis. All measurements were expressed as the mean ± standard error of the mean (SEM). Alterations in groups were determined with a one-way analysis of variance (ANOVA), followed by Tukey’s post hoc test and two-way ANOVA by considering Bonferroni post-tests to compare the means of the replicates, where significance level (*p*) was fixed at 0.05.

## 3. Results

### 3.1. Classification Criteria of Seminiferous Epithelium Apoptosis

Morphological changes in seminiferous tubules showed a marked difference in testes’ histological features of the control and experiment groups. In the control group, ST was filled with germ cells at all days of age, i.e., 181, 200, and 227, and spermatogenesis seemed normal, while in INH immunized group, testes histoarchitecture and germ cells seemed abnormal. ST was classified according to the following apoptosis criteria in Yangzhou ganders at 181, 200, and 227 days of age as shown in Figure 1. Sections were blindly evaluated.

Seminiferous epithelium lumen seemed quite empty, showing degeneration of spermatogonia, spermatocytes, and apoptotic bodies.Seminiferous tubules basal membrane was observed empty. Pyknotic germ and Sertoli cells were also observed.Sertoli cells vacuolated, and most tubules had empty lumen depicting impaired spermatogenesis.Seminiferous tubules had irregularly shaped and degenerated germ cells.

### 3.2. Diameter of ST, LD, EH, Number of ST/Field, and Germ Cells

Histological diagrams in Figure 2 show a method of measuring ST diameter, epithelial heights (EH1, EH2) and luminal tubular diameter (LD). Graphs in Figure 2A–H show numbers of germ cells and ST diameter. There is a positive correlation among ST diameter, epithelial heights, and LT diameter in all experimental days in both groups. Similarly, the number of spermatogonia, spermatocytes, and round and elongated spermatids is higher in control group A than the INH immunized group. Particularly on day 227, when the testes are fully matured, germ cell numbers (spermatogonia and spermatids) are significantly higher in group A than group B, while ST/field is inversely proportional to other histological measurements.

### 3.3. Johnson’s Score

The above mentioned criteria depicted Johnson’s score. On days 181, 200, and 227, Johnson’s score was highest in the control group, showing normal spermatogenesis and germ cells development, while this was not the case in INH immunized group. In group B, disruption in germ cells development and apoptosis resulted in a lower Johnson’s score. Figure 3 shows empty lumen and fewer germ cells in the INH group on all days. Graphical representation of mean Johnson’s score is also shown, depicting a higher mean Johnson’s score in the control than the INH group.

## 4. Discussion

To our knowledge, the present study is the first to illustrate apoptosis caused by inhibin immunization in testicular histoarchitecture of Yangzhou ganders. For normal spermatogenesis, high fertility, and semen quality, normal development of germ cells (spermatogonia, spermatocytes, and spermatids) is required [15]. Immunization against inhibin (INH) improved reproductive efficiency and testes weight of rams [17]. As mammals and birds have less marked morphological differences in the seminiferous epithelium, cellular association occupies a minor area, and spermiogenesis duration is also shorter [18]. In this aspect, our results are per [19], which concluded that INH lowers spermatogonial numbers in testes of adult mice and Chinese hamsters. In our results, numbers of spermatogonia started decreasing after INH immunization. The number of spermatogonia in the experiment group can be explained by the accentuated role of INH [19]. The presence of ascending numbers of apoptotic cells in the experiment group enhances importance for normal spermatogenesis.

Apoptosis is a normal physiological phenomenon during testis steroidogenesis, and in seasonal breeder birds, seasonality in breeding enhances apoptosis rate [20,21,22,23]. In our previous findings, immunization against INH α subunit in Yangzhou ganders elevated anti-INH antibody titer after each INH followed by booster immunizations on 181, 200, and 227 days of age [10]. From these findings, we can observe testis atrophy, resulting in apoptotic testicular histoarchitecture after each INH immunization. These two concomitant phenomena seem to be linked, as the anti-INH antibody tire is low in the control group with normal spermatogenesis.

However, only INH immunization does not seem to be the single reason for apoptosis in testis sections. In seasonal breeders, seasonality in breeding is one of the most important factors affecting testes steroidogenesis. In seasonal breeding geese, photoperiod has an effect and correlation with gonadal activity [24,25]. Apoptosis is a normal phenomenon in proliferating and continuous rejuvenating tissues and seminiferous epithelium [26]. In seasonal breeders, seasonality also effects testis development. Photoperiod effects gonadotrophins [24,25] and testis development, which correlates with the breeding season in geese [27]. Serum INH-B levels in the body vary throughout life, and its regulation deviates at puberty. INH-B regulation and production are mediated by FSH and Sertoli cells before puberty, while after maturity it is modulated by germ cells [28]. Yangzhou ganders reach sexual maturity at 227 days of age [15]. After reaching adulthood, the absence of germ cells due to apoptosis may disrupt steroidogenesis of INH-B secretion in matured testis, causing an imbalance between INH levels for normal functioning of Sertoli and Leydig cells.

Leydig cells play a pivotal role in spermatogenesis regulation [29]. Somatic cells, including Leydig and Sertoli cells, are essential for the normal regulation of spermatogenesis [30]. Ultimately, spermatogenesis is disturbed. To maintain spermatogenesis, INH is crosslinked between Sertoli and Leydig cells [10]. Yangzhou geese start egg-laying in autumn, peak between February and March, and end in May and June [27]. Therefore, it is evident that reproductive seasonality in Yangzhou ganders ends in May; our experiment started at the end of April and ended in June. At that specific timing of seasonality, the breeding season is almost over in Yangzhou ganders, and concomitant effect of INH and seasonality may be the reason for testicular apoptosis.

Testicular histology varies with breeding stage and age of maturity. As birds move from premature to transition and another mature testis, various seminiferous epithelium waves cause alterations in testis histology. During the breeding season, testes size is positively correlated with sperm production [31]. In our previous study, INH immunization in Yangzhou ganders elevated testes weight due to Sertoli cells specific genes. The Sertoli cell is a central regulator of testes development [10]. Johnsen’s score was lower in the INH group and highest in the control group, having a perfect state of spermatogenesis [15]. Several spermatogonia, spermatocytes, round and elongated spermatids, EH, and diameter ST are positively correlated, depicting that Yangzhou ganders moved from premature to mature testis [32]. Elevation in the germ cell numbers in the seminiferous epithelium is related to the ST diameter, LD, and EH [15]. ST/field lowered with the age of maturity. ST/field is inversely proportional to testes weight, ST diameter, EH, and LD [15]. Measuring Sertoli cell numbers was beyond the scope of the present study.

## 5. Conclusions

Inhibin immunization disrupted germ cells and testicular histology, showing apoptosis and impaired spermatogenesis. INH immunization lowered the efficiency of spermatogenesis in Yangzhou ganders. The concomitant effect of INH immunization and seasonality seemed to be the reason for apoptotic bodies in testis. However, imbalanced endogenous levels may be a reason for disrupted seminiferous epithelium. Molecular mechanisms and apoptotic pathways are needed to explore apoptosis by INH immunization.

## Figures and Tables

**Figure 1 animals-11-02801-f001:**
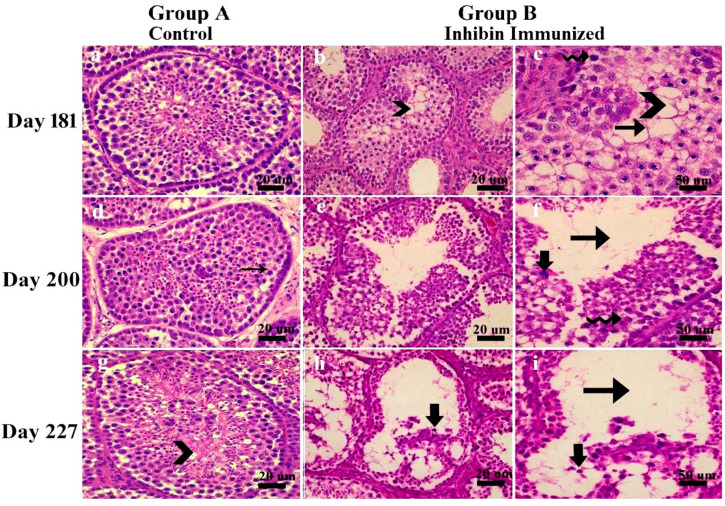
Histological pictures showing apoptosis of seminiferous epithelium in testes of Yangzhou ganders at 181, 200 and 227 days of age in INH immunized group. (**a**,**d**,**g**) belong to control group while (**b**,**c**,**e**,**f**,**h**,**i**) were taken from INH group. Scale bar is 20 µm and 50 µm at 40×. Black arrow with tail: Empty lumen, Spiral arrow: degenerated spermatogonia, Chevron: Sertoli cell vacuolation, Vertical arrow: Apoptotic bodies, Small thin arrow: Spermatogonia. Each group *n* = 30.

**Figure 2 animals-11-02801-f002:**
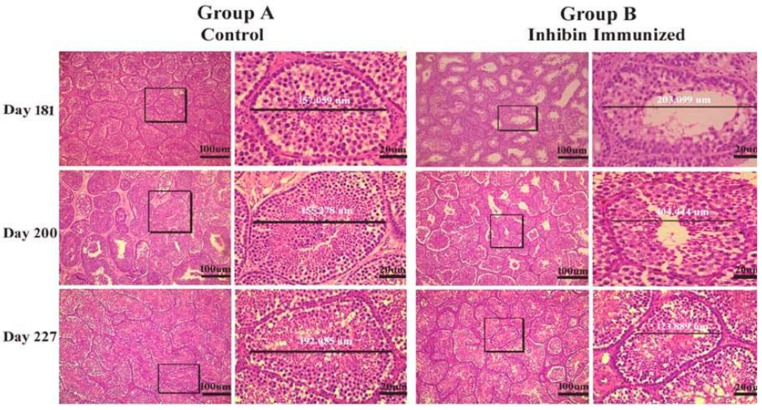
Determination of histological measurements of seminiferous tubules in Yangzhou ganders. TD-tubular diameter, LD-luminal tubular diameter determined by difference between TD and sum of epithelial heights (EH1+EH2). Epithelial height determined by mean of EH1 and EH2, scale bar is 100 μm and 20 μm at 10× and 40× magnification respectively. Subfigures (**A**–**H**) depicts morphometric measurements of histological sections and germ cells numbers. Data are shown as the mean values ± standard error of the mean, ** and *** indicates statistical significance based on *p* < 0.01 and *p* < 0.001, respectively between group A (control) and group B (INH immunized). Each group *n* = 30.

**Figure 3 animals-11-02801-f003:**
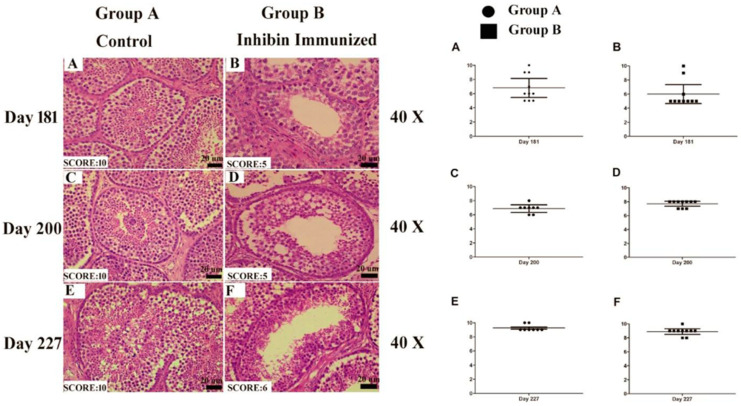
Images of histological sections on days 181 (**A**,**B**), 200 (**C**,**D**) and 227 (**E**,**F**) showing seminiferous tubules at different stages of spermatogenesis by the Johnsen score. Scale bar 20 μm at 40× magnification. (**A**–**F**) Graphical representation of the Johnsen score on days 181, 200 and 227 at a 95% confidence interval. Scale bar = 20 μm. Each group *n* = 30. ● Group A (Control) and ▪ Group B (Inhibin Immunized). Dots and squraes in Graphs (**A**–**F**) representing 10 birds each in both groups while Y axis shows Jonhsons score (0–10).

## Data Availability

Not applicable.

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
