# Peer review of "The Effect of Inhibin Immunization in Seminiferous Epithelium of Yangzhou Goose Ganders: A Histological Study"

_animals, 2021, doi:10.3390/ani11102801_

Round 1

Reviewer 1 Report

This is a preliminary study that relies only on routine histology whose purpose was to evaluate the effect of inhibin immunization in seminiferous epithelium of Yangzhou goose ganders. The Results suggest that concomitant effect of inhibin immunization and seasonality in breeding regressed germ cells and damaged spermatogenesis in seminiferous epithelium.

COMMENT 1: I think it's too risky, too ambitious, to say in the title, with the techniques used, without any marker of apoptosis, to say “Apoptosis mediated testicular regression…” I suggest: “The Effect of Inhibin Immunization in Seminiferous Epithelium of Yangzhou Goose Ganders: A Histological Study”

INTRODUCTION

COMMENT 2: In lines 63-65 I suggest adding “Yangzhou goose ganders”

“In our previous study, active immunization against INH affected spermatogenesis and testicular development through expression of hypothalamic, pituitary and testicular genes in Yangzhou goose ganders"

MATERIALS AND METHODS

COMMENT 3 (line 100): Body weight?

COMMENT 4: (line 100): Why 161 days? Is the reproductive age in this species (say here or in the Introduction).

COMMENT 5 (line 102): Was the immunization always at the same time of day?

COMMENT 6: (line 111): Testicles were not weighed? And the appearance in the experimental group was normal?

COMMENT 7: The authors should articulate sections 2.6, 2.7 and 2.8 both are microscopy, I don't think it makes sense to separate them. One the other hand I refer for all histomorphometric criteria how many evaluations were made per section/animal.

COMMENT 8: What are the criteria for saying the cells were in apoptosis? And was any index determined?

COMMENT 9: Were the observations made blindly?

COMMENT 10: How the photomicrographs were obtained?

RESULTS

COMMENT 11: As in the other photomicrographs, put "Control" and Inhibin “Immunized” in Fig 1.

COMMENT 12: The photomicrographs do not allow us to observe some of the findings to which the authors refer, for example, apoptotic cells, vacuolization of Sertoli cells...Please increase the quality/definition

COMMENT 13: In the figures/graphs please, put the N

DISCUSSION

COMMENT 14: The authors refer that “Apoptosis is normal physiological phenomenon during steriodogenesis of testis and in seasonal breeder birds, seasonality in breeding enhances rate of apoptosis [22-25]”. Without using apoptosis markers (TUNEL, caspases, etc) it is difficult to say, without having quantified that apoptosis mediated testicular regression induced by inhibin immunization.

COMMENT 15: Since the authors base the Discussion on apoptosis and assuming that it is primarily responsible for the observed changes, why did they not evaluate the expression of apoptosis markers?

COMMENT 16: Because of the relationship that exists between the two types of cells, why did they not observe any changes in Leydig cells?

Author Response

COMMENT 1: I think it's too risky, too ambitious, to say in the title, with the techniques used, without any marker of apoptosis, to say “Apoptosis mediated testicular regression…” I suggest: “The Effect of Inhibin Immunization in Seminiferous Epithelium of Yangzhou Goose Ganders: A Histological Study”

Response: Corrected as suggested

INTRODUCTION

COMMENT 2: In lines 63-65 I suggest adding “Yangzhou goose ganders”

“In our previous study, active immunization against INH affected spermatogenesis and testicular development through expression of hypothalamic, pituitary and testicular genes in Yangzhou goose ganders"

Response: Corrected as suggested. Line 70 in updated version.

MATERIALS AND METHODS

COMMENT 3 (line 100): Body weight?

Response: Line 107 in updated manuscript. Body weight of 4.70 Kilograms (added in text)

COMMENT 4: (line 100): Why 161 days? Is the reproductive age in this species (say here or in the Introduction).

Response: (Added in text Line 108 in updated version) In Yangzhou ganders body maturity is different from sexual maturity i.e. body matures at 161 days of age while sexual maturity is acheived at 227 days of age.

COMMENT 5 (line 102): Was the immunization always at the same time of day?

Response: Yes, all three Inhibin immunizatios at 161, 181 and 227 days of age were done at same time of day.i.e. in morning 09:00 a.m according to China standard time . (UTC+8:00)

COMMENT 6: (line 111): Testicles were not weighed? And the appearance in the experimental group was normal?

Response: Testicles were weight at 161, 181 and 227 days of age and testis weights are already cited in previous published work related to present work i.e.

The role of active immunization against inhibin α-subunit on testicular development, testosterone concentration and relevant genes expressions in testis, hypothalamus and pituitary glands in Yangzhou goose ganders. Theriogenology 128 (2019) 122-132

Testis histological sections in control group showed normal germs cells development in seminiferous epithelium as shown in Figure 1 at 161, 181 and 227 days of age in present work.

COMMENT 7: The authors should articulate sections 2.6, 2.7 and 2.8 both are microscopy, I don't think it makes sense to separate them. One the other hand I refer for all histomorphometric criteria how many evaluations were made per section/animal.

Response: Sections 2.6, 2.7 and 2.8 are articulated as Microscopy. 10 ST/bird were observed for  histomorphometric criteria and for mean Johnson’s score

COMMENT 8: What are the criteria for saying the cells were in apoptosis? And was any index determined?

Response: As this study aimed at histological alterations, critera for germ cells apoptosis was based on morphological changes and observing abnormal and irregular development i.e. pyknotic, degenerated and necrotic shaped germ cells under microscope at high magnification 40X. These irregular morphological observations were present in inhibin immunized group. While in Control group germ cells development was normal.

COMMENT 9: Were the observations made blindly?

Response: Observations were carried by seeing abnormal and normal germ cells development under microscope

COMMENT 10: How the photomicrographs were obtained?

Response: All photomicrographs were obtained by randomly selecting some sections from both groups showing normal and abnormalities

RESULTS

COMMENT 11: As in the other photomicrographs, put "Control" and Inhibin “Immunized” in Fig 1.

Response: Corrected as suggested

COMMENT 12: The photomicrographs do not allow us to observe some of the findings to which the authors refer, for example, apoptotic cells, vacuolization of Sertoli cells...Please increase the quality/definition

Response: Corrected as suggested

COMMENT 13: In the figures/graphs please, put the N

Response: Added Each group N=30 in      figures/graphs

DISCUSSION

COMMENT 14: The authors refer that “Apoptosis is normal physiological phenomenon during steriodogenesis of testis and in seasonal breeder birds, seasonality in breeding enhances rate of apoptosis [22-25]”. Without using apoptosis markers (TUNEL, caspases, etc) it is difficult to say, without having quantified that apoptosis mediated testicular regression induced by inhibin immunization.

Response: In this study, changes in seminiferous epithelium and germ cells induced by inhibin was purely based on (H&E) histological study. So changes in nucleus (cracked into two or apoptotic bodies) cytoplasm intensely eosinophilic shows apoptosis signs. In this aspect, apoptosis was observed in histological sections of Inhibin immunized group as shown in pictomicrographs.

COMMENT 15: Since the authors base the Discussion on apoptosis and assuming that it is primarily responsible for the observed changes, why did they not evaluate the expression of apoptosis markers?

Response: Present study was aimed to observed changes in testicular histological pictures based on H&E. Thats why apoptotic markers were not evaluated

COMMENT 16: Because of the relationship that exists between the two types of cells, why did they not observe any changes in Leydig cells?

Response: Present study was focused on propagation of spermatogenesis based on Sertoli, spermatogonia, spermatocytes and seminiferous epithelium apoptosis. Leydig cells were not observed because they produce testosterone while Sertoli cells are mainly responsible for testis size enhancement and nourishment of germ cells

Reviewer 2 Report

Considerable linguistic refurbishment is required for the reader to be able to follow the results and discussion sections. The data are good, but presentation does not do credit to them. Some of the pictures are not sharp. The fixative did not do justice to the tissue structure. Bouin's fluid is an excellent fixative that does not cause too many fixation artefacts.

Author Response

Considerable linguistic refurbishment is required for the reader to be able to follow the results and discussion sections. The data are good, but presentation does not do credit to them. Some of the pictures are not sharp. The fixative did not do justice to the tissue structure. Bouin's fluid is an excellent fixative that does not cause too many fixation artefacts.

Response: Quality of figures 1 and 3 are tried to be improved. The Author has tried to improve quality of research paper in terms of English language in Results and Discussion section.

               Before Correction

             After Correction

RESULTS

1. Line 176

Features in control and experimental group changed to

 RESULTS

Features of control and experimental group

2. Line 176

In control group, at all days 181, 200 and 227 changed to 

In control group, at ll days of age i.e. 181, 200 and 227

3. Line 178

INH immunized group, testes histoarchitecture didn't seemed normal and germ cells were observed damaged , changed to

 INH immunized group, testes histoarchitecture and germ cells seemed abnormal

4. Line 176

Seeing morphological changes in testes

      Deleted

5. Line 189

Histological diagrams showing methods of measuring ST diameter, Epithelial heights (EH1, EH2) and luminal tubular diameter (LD). Figure 2 (A-H) shows histological measurements

Changed to

Histological diagrams in figure 2  shows method of measuring ST diameter, Epithelial heights (EH1, EH2) and luminal tubular diameter (LD). Graphs in figure 2 (A-H) shows numbers of germ cells and ST diameter

 6.  Line 200

Deleted

Propagation of

7. Line 201

Disruption in germ cells development and apoptosis resulted in lower Johnson’s score in group B

Changed to

In group B disruption in germ cells development and apoptosis resulted in lower Johnson’s score

8. Line 209

Added

and semen quality

9. Line 210

Deleted

to attain high fertility and high semen quality

10. Line 211

Immunization against inhibin improved reproductive efficiency of rams and improved testes weight

Changed to

Immunization against inhibin improved reproductive effeciency and testes weight of rams

11. Line 212

In birds, there are not marked morphological differences in seminiferous epithelium as compared to mammals, though cellular associations occupies minor area in ST and spermatogenesis duration is shorter

Changed to

As compared to mammals, birds have less marked morphological differences in seminiferous epithelium, cellular association occupies minor area and spermiogenesis duration is also shorter

12. Line 218

Added

In our results

13. Line 230

Added in start of sentence

In seasonal breeders

14. Line 231

Added in start of sentence

In seasonal breeding in geese

Reviewer 3 Report

The authors showed the effect of inhibin immunization in the seminiferous epithelium in Yangzhou goose ganders. They described regressed germ cells, damaged spermatogenesis and apoptosis in the treated group. The results are well presented, and the methodology applied is suitable for the proposed objectives. While the data is original, it is an expansion of the study performed in a previous paper (reference 10). However, it is relevant information therefore recommend accepting this manuscript in its present form.

I found two typos:

Line 51, it should say “Sertoli” not “Setoli”

Line 246, it should say “disturbed” not “disturebed”

Author Response

The authors showed the effect of inhibin immunization in the seminiferous epithelium in Yangzhou goose ganders. They described regressed germ cells, damaged spermatogenesis and apoptosis in the treated group. The results are well presented, and the methodology applied is suitable for the proposed objectives. While the data is original, it is an expansion of the study performed in a previous paper (reference 10). However, it is relevant information therefore recommend accepting this manuscript in its present form.

Response: Thanks alot

Round 2

Reviewer 1 Report

The authors adequately answered my questions. However they just didn't understand questions 9 and 10.

COMMENT 9: Were the observations made blindly?
Response: Observations were carried by seeing abnormal and normal germ cells development under microscope.
What I intend to know is whether the investigator who made the observations knew (or not) the origin of the slides to avoid bias. It would be interesting to write "...sections were blindly evaluated...

COMMENT 10: How the photomicrographs were obtained?
Response: All photomicrographs were obtained by randomly selecting some sections from both groups showing normal and abnormalities
What I intend to know is the equipment as the authors refer to the microscope.It would be interesting to write " The photomicrographs had been obtained using a ???? digital camera procedure fitted to the microscope.

Author Response

COMMENT 9: Were the observations made blindly?
Response: Observations were carried by seeing abnormal and normal germ cells development under microscope. What I intend to know is whether the investigator who made the observations knew (or not) the origin of the slides to avoid bias. It would be interesting to write "...sections were blindly evaluated...
Response: added as suggested in Results section. 3.1. Sections were blindly evaluated.
COMMENT 10: How the photomicrographs were obtained?
Response: All photomicrographs were obtained by randomly selecting some sections from both groups showing normal and abnormalities What I intend to know is the equipment as the authors refer to the microscope.It would be interesting to write " The photomicrographs had been obtained using a ???? digital camera procedure fitted to the microscope
Response: added in Microscopy 2.6. Section. The photomicrographs had been obtained using a 8 Mega Pixel CCD digital camera procedure fitted to the microscope.

Reviewer 2 Report

This manuscript has some good material to report, but it cannot be understood from the language point of view. It needs complete re-writing in good English.

Author Response

Author has tried to improve english language in updated manuscript version.
